# Peer review of "The Multi-Station Fusion-Based Radiation Source Localization Method Based on Spectrum Energy"

_sensors, 2025, doi:10.3390/s25051339_

Round 1

Reviewer 1 Report

Comments and Suggestions for Authors

This paper presents a multi-station fusion-based method for radiation source localization, which utilizes characteristic information such as frequency, field strength, bandwidth, and other parameters embedded in frequency band scanning data. However, the authors introduced the proposed method but did not compare the results with other similar methods. Moreover, there is relatively little analysis in this article, and the content is somewhat insufficient. The specific issues are as follows:

1. The objective of this article needs to be clearly stated at the end of the introduction section.

2. The author mentioned that the experiment was conducted in a flat and open plain area without mountains or high-rise buildings. Can the authors provide photos of the relevant scenes and equipment for the experiment?

3.The analysis in section 3.2 is too simplistic, and the author simply compared the calculated results with the measured results. Can the author analyze the advantages of the results obtained by this method in more depth? It is best to quantify the results and compare them with other similar methods currently available.

4. The resolution of Figure 4 needs to be improved.

Author Response

Thank you very much for taking the time to review this manuscript. The detailed responses to your comments and revisions are in the uploaded Word file . We appreciate your valuable feedback, which has helped improve the quality of our work. Please note that we have addressed each of your comments carefully. If there are any points where we respectfully disagree, we have provided explanations for our perspective.

Reviewer 2 Report

Comments and Suggestions for Authors

The authors present an interesting solution to a real-world problem. Without a doubt, the detection of rogue/jamming transmitters is a serious concern and can only be expected to become more important in an increasingly interconnected society where IoT, wireless sensor systems, and autonomous devices play a ever more significant role. It is with that background in mind that I read the paper with much interest and curiosity for the approach the authors have taken to address this problem.

In the introduction, it is suggested that sensor fusion systems can take advantage of multiple different characteristics of RF transmitters such as the field strength, frequency, and bandwidth. However, the manuscript does not adequately explain how each of these parameters are measured and how they are combined to progress towards a localisation solution. Having 3 beacons is a logical choice for localising a point in a 3 dimensional space. However, it appears from the graphs that the only data recorded is the RSSI in the frequency domain 30 MHz to 3 GHz. In this case, it is purely a triangulation problem similar to GNSS systems. It is not explained how for example the bandwidth of the transmitter is affecting the field strength measured at each of the 3 points.

It would be very helpful if a diagram were included, showing the locations of each of the transmitters on a satellite image to give readers a impression of the physical localisation and relative positioning of the beacons. Information on the instrumentation used in the experiments, for example the antennas, amplifiers, and RF FFT analysers, is absent from the manuscript and should be added. The triangulation assumes a perfectly isotropic source, but in the real world, rogue transmitters are unlikely to use antennas with good isotropic characteristics. This is an issue that should be addressed in the discussion section. For an isotropic emission, I expected to see the inverse square law in the theoretical model, its absence deserves further explanation.

To demonstrate the viability of the presented approach, I would suggest to add non-linearities to the problem. Without a doubt, there is scientific agreement that triangulation of measured field strength works in theory, but it would be most useful to do additional experiments for example varying distances between antennas, introducing objects that cause reflections, and non-isotropic antennas.

Author Response

(The authors gave the same response as above.)

Round 2

Reviewer 1 Report

Comments and Suggestions for Authors

The authors have made corresponding modifications and the quality of the article has been improved, and it can now be published.